# Electrically insulating PBO/MXene film with superior thermal conductivity, mechanical properties, thermal stability, and flame retardancy

Yong Liu[1,2], Weizhi Zou[1,2], Ning Zhao [1,2] ✉ & Jian Xu [1]

Constructing flexible and robust thermally conductive but electrically insulating composite films for efficient and safe thermal management has always been a sought-after research topic. Herein, a nacre-inspired high-performance poly(p-phenylene-2,6-benzobisoxazole) (PBO)/MXene nanocomposite film was prepared by a sol-gel-film conversion method with a homogeneous gelation process. Because of the as-formed optimized brick and mortar structure, and the strong bridging and caging effects of the fine PBO nanofibre network on the MXene nanosheets, the resulting nanocomposite film is electrically insulating ($2.5 \times 10^9\,\Omega\,cm$), and exhibits excellent mechanical properties (tensile strength of 416.7 MPa, Young's modulus of 9.1 GPa and toughness of $97.3\,MJ\,m^{-3}$). More importantly, the synergistic orientation of PBO nanofibres and MXene nanosheets endows the film with an in-plane thermal conductivity of $42.2\,W\,m^{-1}\,K^{-1}$. The film also exhibits excellent thermal stability and flame retardancy. This work broadens the ideas for preparing high-performance thermally conductive but electrically insulating composites.

Electronic devices are developing towards miniaturization, integration, and intelligence[1,2]. Microelectronics with high power levels often generate excessive heat, which negatively affects the performance of the products and can also cause unexpected fire incidents[3]. To address this urgent issue, polymer-based thermal conductive materials have attracted widespread attention due to their ease of processing, flexibility, and light weight[4–6]. High thermal conductive nanofillers such as metal nanoparticles, graphene nanosheets (GNSs), carbon nanotubes (CNTs) and boron nitride nanosheets (BNNSs) are essential in the manufacturing of polymer-based thermal conductive materials because of the thermal insulation of common polymer ($<0.5\,W\,m^{-1}\,K^{-1}$)[7–9]. Although metal particles, GNSs and CNTs can form effective thermal conductive pathways in composites, their high electrical conductivity cannot satisfy the electrical insulation requirements of electronic devices. Therefore, polymer/BNNS thermal conductive composites with excellent electrical insulation are

more suitable for applications in electronic devices. However, high-quality BNNSs are not easy to prepare due to the strong interlayer forces in BN bulks, and the inert surface of BNNS makes it difficult to achieve both high thermal conductivity and excellent mechanical properties in the composites[8–10].

Ultra-thin $Ti_3C_2T_x$ MXene nanosheet, an emerging two-dimensional (2D) material with abundant surface functional groups, can be acquired by exfoliating $Ti_3AlC_2$ MAX phase, and has been employed in various applications including photocatalysis, sensors, electromagnetic interference shielding, etc.[11]. Recently, MXene has shown notable potential for thermal management since the theoretical thermal conductivity of monolayer $Ti_3C_2T_z$ can be as high as $108\,W\,m^{-1}\,K^{-1}$[12]. Regrettably, freestanding MXene films exhibit poor mechanical properties[13–15]. Composites constructed from 1D polymer nanofibres and 2D MXene nanosheets by mimicking the natural nacre structure have been demonstrated to break through the barrier in

[1]Beijing National Laboratory for Molecular Sciences, Laboratory of Polymer Physics and Chemistry, Institute of Chemistry, Chinese Academy of Sciences, Beijing, PR China. [2]University of Chinese Academy of Sciences, Beijing, PR China. ✉e-mail: zhaoning@iccas.ac.cn

mechanical properties[16–22]. For example, nacre-like composite films of MXene/aramid nanofibre, MXene/nanocellulose and MXene/bacterial cellulose nanofibre have shown a tensile strength of 198.8, 416.1, and 297.5 MPa respectively, far greater than those of the pristine MXene films[20–22]. However, this kind of composite still has conflicting electrical insulation and thermal conductivity because of the continuous conductive paths established by MXene nanosheets. To date, electrical resistance above $1 \times 10^9 \, \Omega \, cm$ and thermal conductivity exceeding $10 \, W \, m^{-1} \, K^{-1}$ have yet to be realized in MXene-based composites (Supplementary Tables 1 and 5).

Herein, nacre-like composite film of poly(p-phenylene-2,6-benzobisoxazole) (PBO) nanofibres and MXene nanosheets was fabricated through a sol-gel-film conversion approach. PBO nanofibre was chosen because of its remarkable mechanical properties, thermal stability, fire safety and thermal conductivity[23–26]. Homogeneous gelation was achieved to improve the fineness and robustness of the PBO nanofibre network and reduce the aggregation of MXene nanosheets. Because of the optimized brick and mortar structure and the intrinsic nature of the building blocks, the nanocomposite film simultaneously achieves high thermal conductivity and desirable electrical insulation, which has not been reported before. Meanwhile, a combination of superior strength and toughness, and outstanding thermal stability and flame retardancy was also realized. This high-performance thermally conductive, yet electrically insulating film makes it an ideal material for managing electronic device heat in a safe and effective manner.

## Results

### Fabrication strategy and characterization of PBO/MXene nanocomposite film

Figure 1a illustrates the schematic preparation of $Ti_3C_2T_x$ MXene nanosheets by etching and exfoliating $Ti_3AlC_2$ MAX phase. MXene nanosheets of approximately 1.7 nm thickness have been obtained (Fig. 1b and Supplementary Fig. 1a, b). X-ray diffraction (XRD) investigations show that the (002) peak of MAX at 9.51° shifts to 6.71° after the exfoliation, confirming the successful preparation of MXene (Supplementary Fig. 1c)[27]. The preparation of PBO nanofibres is schematically shown in Fig. 1c. Commercial PBO fibres (diameter of approximately 9.7 μm) were protonated and exfoliated into PBO nanofibres in the presence of methane sulfonic acid (MSA) and trifluoroacetic acid (TFA) (Fig. 1d). The average diameter of PBO nanofibres is 17 ± 1 nm according to the transmission electron microscopy (TEM) image.

The nacre-inspired PBO/MXene nanocomposite film was produced through the sol-gel-film conversion approach including a proton-consumption homogeneous gelation process (Fig. 1e). PBO/MXene sol was obtained by mixing the dispersions of PBO nanofibres and MXene nanosheets. Then MSA/Ethyl acetate (EA)/$H_2O$ was added to trigger the gelation of PBO/MXene sol. During the gelation process, EA was progressively disassembled towards ethanol and acetic acid, and ethanol not only accepted protons of MSA, TFA, and protonated PBO, but also consumed TFA via an esterification reaction. The uniformly deprotonated PBO nanofibres are critical for the construction of the fine 3D network in the gel[25]. The PBO/MXene alcohol gel was obtained by replacing the solvents in the acid gel with isopropanol. The storage modulus (G′) of the alcohol gels is one order of magnitude higher than its loss modulus (G″), confirming the formation of a robust 3D gel network (Supplementary Fig. 2). Finally, the alcohol gel was dried in air to obtain the nacre-like nanocomposite film with PBO nanofibres and MXene nanosheets horizontally oriented. The nanocomposite films with MXene content of 10, 20, 30, 50, and 70 wt% were named PM10, PM20, PM30, PM50, and PM70, respectively.

To better understand the as-formed 3D network in the gel, the gel was freeze-dried to largely retain the porous structure. SEM image indicates that the 3D network of the PBO gel is constructed from uniformly dispersed fine nanofibres (Supplementary Fig. 3a). Under the capillary force, the 3D nanofibre network is transformed into a dense film with a laminated nature (Supplementary Fig. 3b). For PBO/MXene gel, MXene nanosheets are homogeneously distributed in the PBO nanofibre network, and PBO/MXene films inherit similar laminated structures (Fig. 1f–i and Supplementary Fig. 4). The energy dispersive spectroscopy (EDS) mapping images also confirm the homogeneity of the MXene distribution in the laminar composite film (Supplementary Fig. 5).

It is worth mentioning that the isopropanol we used is less able to accept protons than water[28], which can reduce the uneven aggregations of nanofibres during solvent replacement, and isopropanol can also retard the oxidation of MXene very well[29]. In addition, the high free volume due to the dilution effect of MSA/EA/$H_2O$ on PBO/MXene sol and the small linear shrinkage rate (~8%) during the solvent exchanging process is beneficial for reducing aggregation between MXene nanosheets. Thus, the PBO/MXene films have an optimized brick and mortar network structure. As shown in Supplementary Fig. 6, the average nanofibre diameter in the PM50 gel (20 ± 1 nm) is much smaller than that in the control sample of PBO/MXene composite gel with 50 wt% MXene content prepared by water-vapor-induced gelation (named W-PM50 gel, 164 ± 3 nm)[30], and a finer and denser nanofibre network adheres to the MXene nanosheets in the PM50 gel. Meanwhile, severe aggregation of MXene nanosheets in the W-PM50 gel network can be also observed.

XRD patterns of the films indicate the characteristic diffraction peak (002) attributed to MXene decreases from 6.71° to 6.35° as the PBO content increases, suggesting the crystal plane spacing of MXene nanosheets is gradually larger (Fig. 1j). This change confirms that the PBO nanofibres are successfully inserted into the MXene interlayer. X-ray photoelectron spectroscopy (XPS) spectra show that Ti and F of MXene are almost undetectable on the surface of the composite films (PM20 and PM50, Fig. 1k and Supplementary Table 2), revealing the caging of the PBO nanofibres on the MXene nanosheets. In addition, many polar functional groups (−F, −O and −OH) can be found on the MXene film surface (Fig. 1k and Supplementary Fig. 7a)[11]. Polar groups (−C = O and −C − N) can be also noticed in the high-resolution C 1 s spectrum of PBO film because of the partial degradation during the exfoliation of the PBO fibres (Supplementary Fig. 7b)[31]. Fourier transform infrared spectroscopy (FTIR) spectra confirm that strong hydrogen bonds are formed between the units, since the C − C and C = C vibration absorptions of the benzene ring at 1494 and 1576 $cm^{-1}$, and the C = N and C − O − C stretching vibrations of the oxazole ring at 1618 and 1051 $cm^{-1}$ of PBO shift to a lower wavenumber for the PBO/MXene film (Fig. 1l)[32,33]. All the described results reveal that PBO/MXene composite film featuring a nacre-like structure has been formed, in which uniformly distributed MXene nanosheets are bridged and caged by the fine 3D PBO nanofibre network through strong hydrogen bonding interactions. This morphology feature promises not only superior mechanical properties, but also the formation of efficient thermally conductive but electrical insulating pathways.

### Mechanical properties of PBO/MXene nanocomposite film

The stress–strain curves and the detailed mechanical properties of the films with different filler loads are shown in Fig. 2a–c. Because of the reinforcing effect of PBO nanofibre network, the mechanical properties of the composite films are far superior to those of pure MXene film. Moreover, compared with pure PBO film, the mechanical properties of the nanocomposite film are effectively improved at 10% MXene content. The optimal mechanical property values are achieved when the MXene content is 20%, including a tensile strength of 416.7 MPa, a Young's modulus of 9.1 GPa and a toughness of 97.3 $MJ \, m^{-3}$, which are 1.40, 1.54, and 1.63 times those of pure PBO film, respectively. At higher MXene contents, the mechanical properties are gradually reduced,

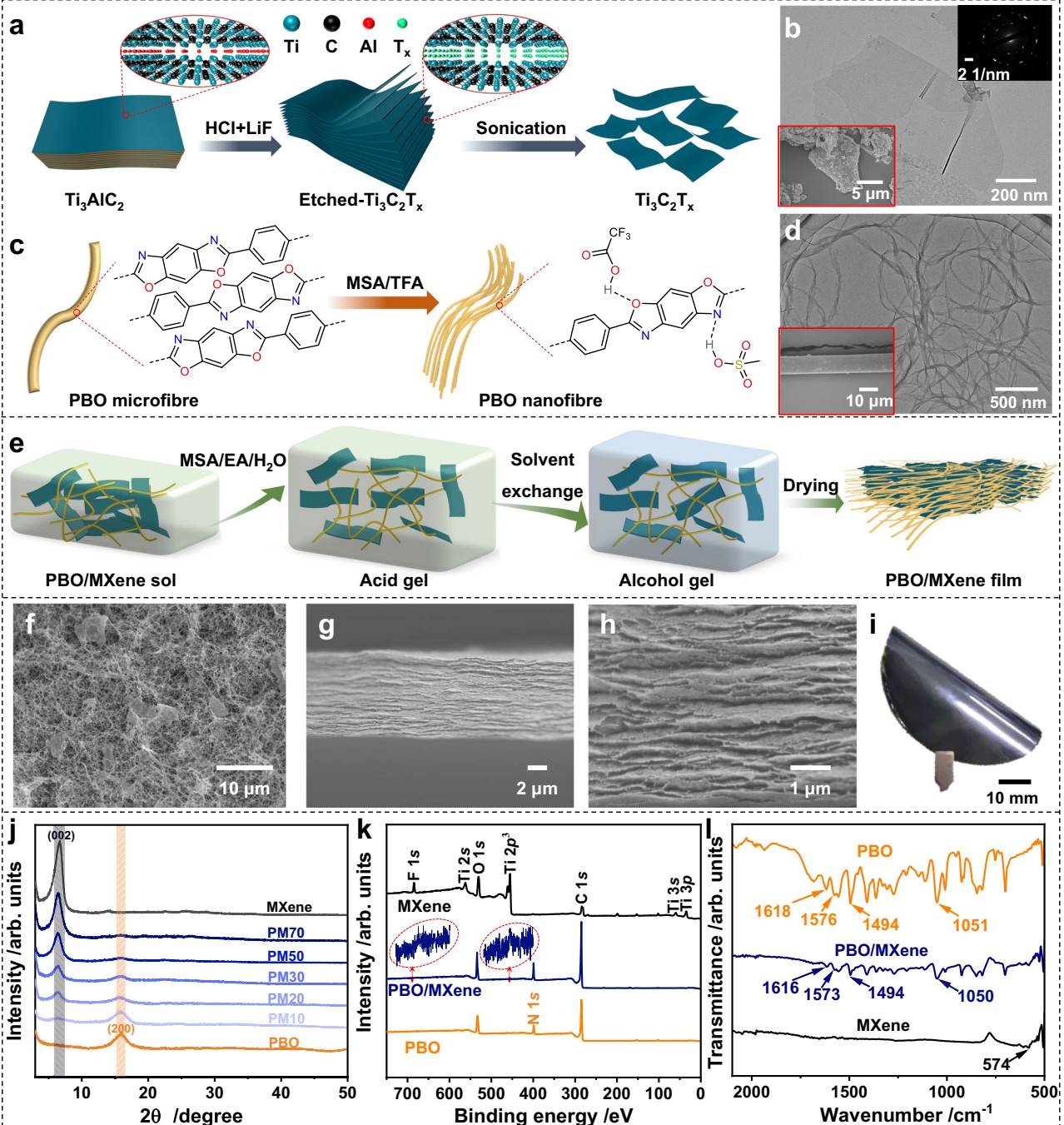

**Fig. 1 | Fabrication and characterization of PBO/MXene nanocomposite films.** **a** Illustration for Ti₃C₂Tₓ MXene prepared from Ti₃AlC₂ MAX precursor by HCl/LiF etching. **b** TEM image of MXene nanosheets (insets: electron diffraction pattern of MXene and SEM image of Ti₃AlC₂ MAX). **c** Illustration for PBO nanofibres prepared from commercial PBO fibres by MSA/TFA exfoliation. **d** TEM image of PBO nanofibres (inset: SEM image of PBO fibre). **e** Illustration for the nacre-inspired PBO/MXene films prepared by the sol-gel-film conversion approach including a proton-consumption homogeneous gelation process. **f**–**h** SEM images of **f** the freeze-dried PBO/MXene gel and **g**, **h** the cross-section of PBO/MXene film. **i** The optical photograph of PBO/MXene film. **j** XRD patterns of various films. **k** XPS spectra of PBO, MXene, and PBO/MXene (PM20) films. The insets are separate high-resolution Ti 2p (449.43–469.43 eV) and F 1s (677.43–697.43 eV) XPS spectra of the PBO/MXene film, respectively. **l** FTIR spectra of PBO, MXene, and PBO/MXene films.

where PM70 is even worse than PBO film. This is because the brick and mortar structure was deteriorated by the aggregation of MXene (Supplementary Figs. 4 and 9).

We further examined the mechanical properties of PM20 film after 10000 cycles of folding-unfolding (Fig. 2d). Except for the decrease in the modulus ($E/E_0 = 0.875$), other properties have a slight variation in comparison to the initial values. Furthermore, the nanocomposite film processes fascinating structural stability against

ultrasonication due to its excellent mechanical properties and hydrophobicity (Supplementary Fig. 8). Generally, strength and toughness are mutually exclusive for most composites. Nevertheless, because of the optimized brick and mortar structure, our PBO/MXene films simultaneously achieve superior tensile strength and toughness compared to the performance of most reported nacre-like composites of polymer/inorganic nanosheet (MXene, BNNS, GNS/rGO, Clay) (Fig. 2e and Supplementary Table 3). Moreover, the as-

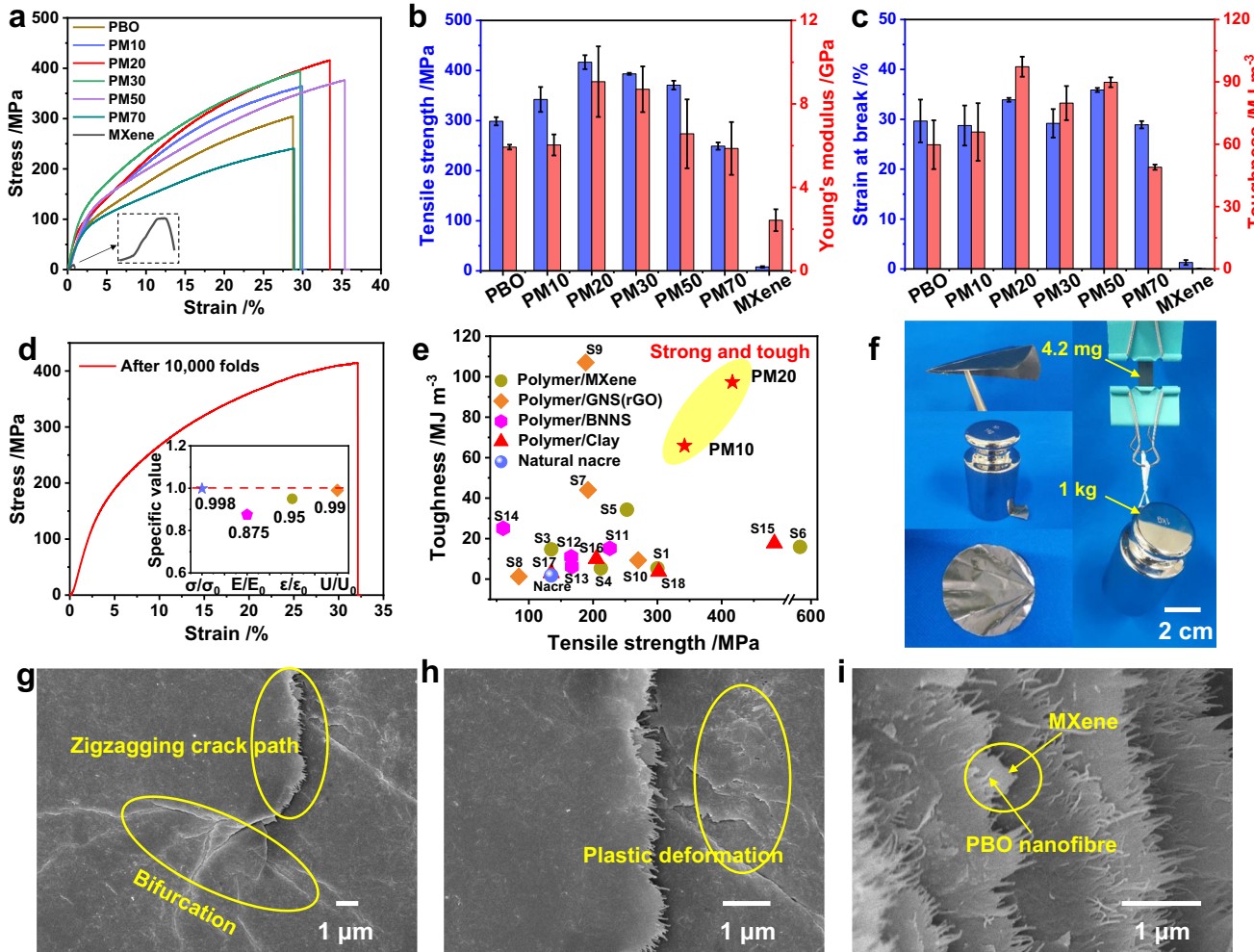

**Fig. 2 | Mechanical properties of PBO/MXene nanocomposite films. a** Typical stress–strain curves of PBO/MXene films with different MXene contents. The magnification factor of inset is 6.35×. **b** Tensile strength and Young's modulus. **c** Strain at break and toughness. The error bars (**b**, **c**) represent the standard deviations of the measured values (*n* = 3). **d** Stress–strain curve of PM20 film after 10000 cycles of folding-unfolding. The inset is tensile strength ratio (σ/σ₀), Young's modulus ratio ($E/E_0$), strain ratio at break ($\varepsilon/\varepsilon_0$) and toughness ratio ($U/U_0$). **e** Comparison of the tensile strength and toughness of PBO/MXene film with those of other nacre-like polymer/2D inorganic nanosheet composites (details shown in Table S3 in the Supplementary Information). **f** The optical photographs displaying the flexibility and robustness of PM20 film. **g–i** SEM images of the propagated crack for PM20 film.

prepared nanocomposite film can be folded into the desired shape, withstand weight pressing, and has no visible damage after unfolding. It can easily lift the weight that is more than 238,000 times its own weight (Fig. 2f).

Fracture morphology has been investigated to reveal the mechanism underlying the strong mechanical properties. As shown in Fig. 2g–i and Supplementary Fig. 9, the typical zigzag crack propagation path, crack bifurcation, and film plastic deformation can be observed during crack propagation in PBO/MXene composite films with 10–50 wt% MXene content, accompanied by slipping and pull out of MXene nanosheets, and elongation and fracture of PBO nanofibres. Notably, more pronounced crack bifurcation and plastic deformation of the surrounding area are shown in the PM20 film compared to the other composite films, indicating a more prominent toughening effect. At MXene contents up to 70 wt%, there is a severe aggregation of nanosheets in the composite film, which could explain the poor mechanical properties even below those of the PBO film. In addition, tensile tests of the single-edge notched films were used to calculate their fracture energy (Supplementary Fig. 10). As expected, PM20 exhibits the greatest fracture energy, further demonstrating its powerful toughening effect on crack propagation. These comparative results indicate that synergistic effects of the optimized nacre-like

nanosheet and nanofibre structure endow the material with strong mechanical properties.

## Thermal conductivity and electrical insulation of PBO/MXene composite film

The thermal conductivities (TCs) of the films were measured using the laser flash method and plotted in Fig. 3a. The detailed parameters (e.g., thermal diffusivity, density, specific heat capacity) are shown in Supplementary Table 4. The in-plane TC of the PBO film is 25.6 W m⁻¹ K⁻¹, outperforming that of most typical polymers mainly due to the high crystallinity of PBO nanofibres and the fine 3D PBO nanofibre network which greatly reduces the phonons scattering[8,34]. The in-plane TCs of the composite films increase with the MXene content first and then decrease. PM20 shows the highest value of 42.2 W m⁻¹ K⁻¹. The remarkably improved TC is closely related to the strong interaction between MXene nanosheets and PBO nanofibres. As a good thermal conductor, MXene nanosheets act as a thermally conductive bridge in the composites, providing a fast phonon/electron channel for heat transfer between PBO nanofibres[35]. At the same time, their unique 2D sheet structure can also effectively improve the orientation alignment of the PBO nanofibres, further increasing the in-plane component of the phonon velocity[36–38]. As shown in Fig. 3b, the two-dimensional

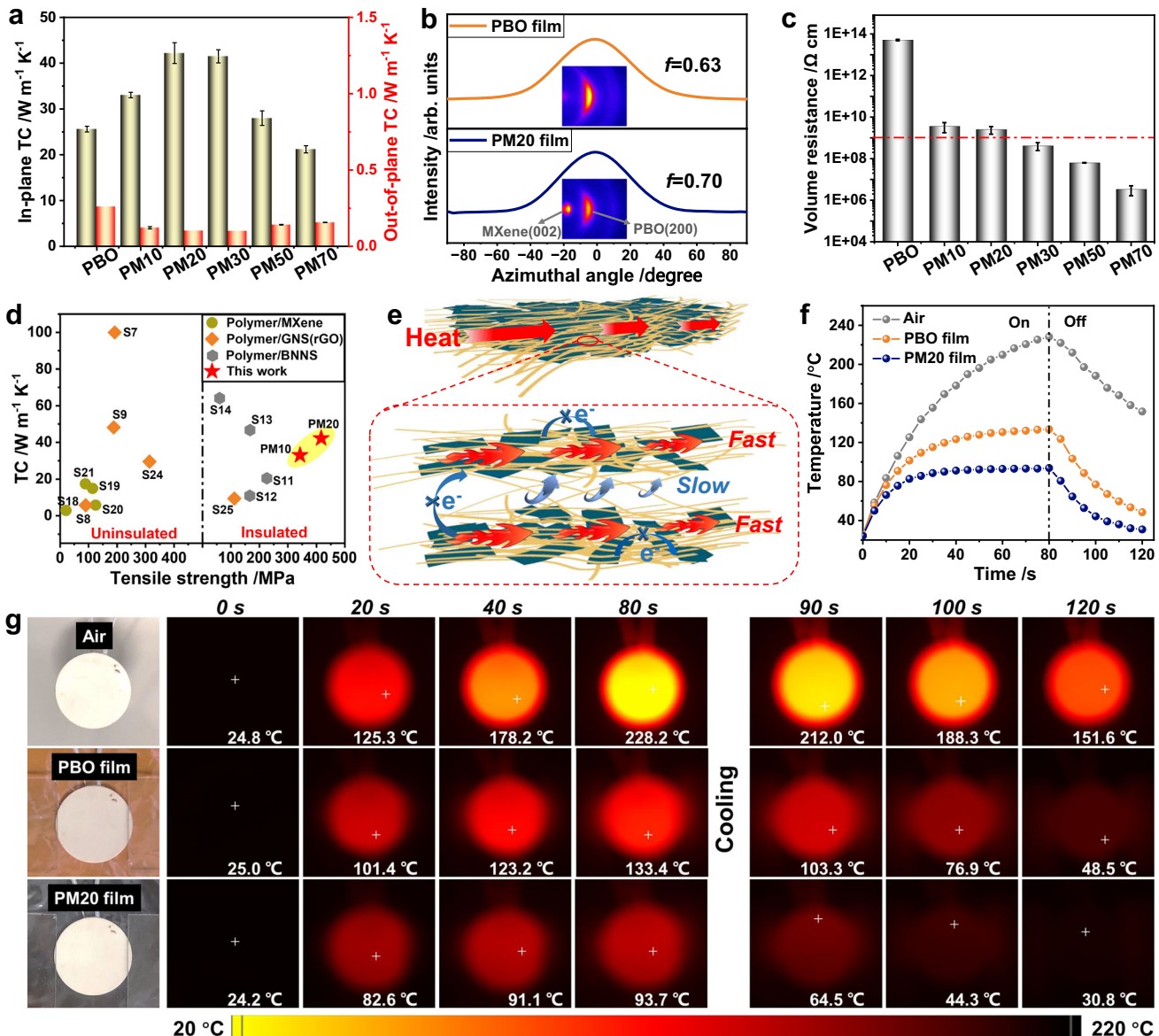

**Fig. 3 | Thermal conductivity and electrical insulation of PBO/MXene nanocomposite films. a** Thermal conductivities (TCs) of PBO/MXene films with different MXene contents. Error bars represent standard deviations. **b** 2D WAXS patterns of PBO and PM20 films and azimuthal profiles for crystal plane (200) reflection of PBO nanofibre. **c** Volume resistance of PBO and PBO/MXene films. The red dashed line represents the critical standard for insulation resistance ($1 \times 10^9 \Omega$ cm). Error bars represent standard deviations. **d** Comparison of the TC and tensile strength of PBO/MXene films and other polymer/2D inorganic nanosheet composites (details shown in Table S5 in the Supplementary Information). **e** Anisotropic thermal conduction and electrical insulation model of PBO/MXene film. **f** Surface temperature–time curves of the ceramic plate using air, PBO film and PM20 film to dissipate heat, respectively. **g** The IR thermal images.

wide-angle X-ray scattering (2D WAXS) pattern of PM20 shows significant anisotropic feature, implying that the PBO/MXene networks are highly oriented along the film face. Notably, Herman's orientation factor (*f*, detailed calculation can be found in the Supplementary Information) in the (200) reflection of PBO nanofibre increases from 0.63 for PBO film to 0.70 for PM20, confirming the enhanced in-plane orientation of the nanofibres with the assistance of the MXene nanosheets. A sharp decrease in the in-plane TC occurs at high MXene content, as the aggregation between MXene nanosheets increases their own thermal resistance and makes it difficult to further improve the orientation alignment of PBO nanofibres. Furthermore, because of this orientation network structure, the TC of the composite film also shows a high degree of anisotropy. The in-plane TC of PM20 is more than 400 times its out-of-plane TC (0.103 W m$^{-1}$ K$^{-1}$, Fig. 3a), which contributes to the efficiency of heat dissipation for local heat sources in practical applications[39].

Excitingly, although the original MXene film has an electrical conductivity of up to 1720 S cm$^{-1}$, the nanocomposite films exhibit extremely high electrical resistance as tested by a high resistance meter (Fig. 3c). In particular, the volume resistances of the PM10 and PM20 films are $3.7 \times 10^9$ and $2.5 \times 10^9 \Omega$ cm, respectively, which already meet the evaluation criteria for electrical insulation ($> 1 \times 10^9 \Omega$ cm). We speculate that such unusually high electrical resistance for the MXene-based composite films can be attributed to two aspects. On the one hand, the dilution effect of MSA/EA/H$_2$O on PBO/MXene sol and the small linear shrinkage of the gel in the horizontal direction for the proton-consumption-induced gelation method make it difficult for the MXene nanosheets to form a conductive pathway at low MXene content. On the other hand, finer and denser PBO nanofibre networks tend to cover the MXene nanosheets more evenly to increase the contact resistance between them. In the control experiment, the electrical conductivity of W-PM50 film is at least three orders of magnitude

higher than that of PM50 film (Supplementary Fig. 11). The morphology differences between W-PM50 and PM50 networks and the distinct electrically conductive behaviors suggest that the structure design has a profound influence on blocking electron transport between MXene nanosheets. The properties of thermally conductive composites of different polymer/inorganic nanosheet have been summarized in Fig. 3d and Supplementary Table 5. Compared to MXene-based thermal conductive materials reported, PM10 and PM20 achieve an unprecedented breakthrough in terms of tensile strength, TC and electrical insulation at the same time. Our composite films also exhibit superior tensile strength and electrical insulation than most GNS/rGO-based thermal conductive materials. Although BNNS-based thermal conductive materials also exhibit excellent electrical insulation, to the best of our knowledge, the tensile strength of our films has surpassed that of the reported BNNS-based thermal conductive materials[10,40–42]. This is mainly due to the inert character of the BNNS surface. These attractive results make our nanocomposite films promising for the thermal management of electronics, for which electrically insulating and thermally conductive materials are preferred.

Figure 3e schematically shows the mechanism of anisotropic heat conduction and electrical insulation of the film. Under strong capillary forces, PBO/MXene gel is transformed into a film with a layered microstructure. The interconnected PBO nanofibres and MXene nanosheets are highly oriented, so heat carriers can be rapidly conducted in the horizontal direction. However, in the vertical direction, heat conduction becomes very slow due to the layered structure with weak interactions (relative to the covalent bonds in the building blocks along the horizontal direction). In addition, the electron conduction between MXene nanosheets is considerably reduced by the barrier of the PBO network.

With the superior in-plane TC of PBO/MXene film in mind, we investigated its effect on dissipating heat from the local heat source (Fig. 3f, g and Supplementary Fig. 12). It should be mentioned that the thickness is much smaller than the diameter for the films (PBO and PM20), and thus the contribution of out-of-plane TC to heat dissipation in the thickness direction is negligible[39,43]. By turning the power supply on/off, the surface temperature of the bare hot plate rises sharply from 24.8 °C to 228.2 °C in 80 s and then drops slowly to 151.6 °C in 40 s. The surface temperature of the hot plate attached to PBO film on the back rises from 25.0 °C to 133.4 °C in 80 s and cools down to 48.5 °C in 40 s. When PM20 film was used, its surface temperature is only 93.7 °C at 80 s and has cooled down to 30.8 °C, close to room temperature, by 120 s. This fact verifies the great potential of PBO/MXene films for high-temperature thermal management in electronic devices.

### Thermal stability and flame retardancy of PBO/MXene nanocomposite film

Thermal stability and flame retardancy of thermal conductive materials are of great value in practical applications. Vertical combustion and limited oxygen index (LOI) tests were carried out to investigate the flame retardancy of the samples[44]. All the films pass the VTM-0 rating (Supplementary Fig. 13) and have a high LOI larger than 50% (Supplementary Fig. 14). Typically, PBO (LOI ~ 53.5%) and PM20 (LOI ~ 54.9%) films exposed to the flame for 10 s self-extinguished immediately after withdrawing from flame, and no molten drippings were produced (Fig. 4a, b). Supplementary Fig. 15a shows the PBO film decomposed almost completely after burning in the flame of alcohol lamp for a long time (120 s), while the composite films turned white partially and the residual size increased with the MXene content, demonstrating an enhanced thermal stability[45]. This is also supported by thermogravimetric analysis (TGA) (Supplementary Fig. 15b).

SEM images show that lots of nanofibres are produced on the surface of burned PBO film and nanoparticles formed on PBO/MXene film, which should have originated from the carbonization of PBO

nanofibres and the oxidation of MXene nanosheets ($TiO_2$), respectively (Fig. 4c, d). XRD and Raman investigations prove the composition changes in the films after burning (Fig. 4e–h)[46,47]. The XRD pattern of the burned PM70 sample still shows peaks of MXene and PBO, further confirming that the introduction of MXene can improve the thermal stability and flame retardancy of the material. In the Raman spectra, the $I_D/I_G$ value (2.68) of the burned PM20 is lower than that of the burned PBO film (3.25), revealing the increased graphitization of the residual carbon due to the effect of $TiO_2$[48]. Therefore, the excellent thermal stability and flame retardancy of the nanocomposite film can be explained by the following mechanism (Fig. 4i). For one thing, the high residual carbon originating from PBO nanofibres[49] and nacre-like brick and mortar structure can greatly retard the transfer of heat and oxygen to the interior. For another, MXene can be used as a support for the robustness of the expanded carbon layer at high temperatures, and its oxidation product $TiO_2$ may promote the formation of high-quality carbon with better thermal stability. Overall, the proper introduction of MXene not only significantly improves the TC and mechanical properties of the PBO film while maintaining the electrical insulating properties, but also further enhances the thermal stability and flame retardancy. Thus, the high-performance nanocomposite film integrating two engineered materials, PBO and MXene, have broader prospects for industrial applications.

## Discussion

In summary, we have successfully fabricated high-performance PBO/MXene nanocomposite films by the sol-gel-film conversion approach with a proton-consumption homogeneous gelation process. Benefiting from the fine and robust 3D interconnected PBO/MXene network and the nacre-like hierarchical structure formed, the film with 20 wt% MXene features unprecedented mechanical properties (tensile strength of 416.7 MPa, Young's modulus of 9.1 GPa and toughness of 97.3 MJ m$^{-3}$), in-plane TC (42.2 W m$^{-1}$ K$^{-1}$) and electrical insulation (2.5 × 10$^9$ Ω cm). Combined with excellent thermal stability and flame retardancy, PBO/MXene composite films have great promise for the high-temperature thermal management of flexible electronics. This work indicates that high thermally conductive but electrically insulating composites can be constructed by using electrically conductive fillers through the structure design.

## Methods

### General information

For raw materials, fabrication procedures of Ti$_3$C$_2$T$_x$ MXene nanosheets and PBO nanofibres, and material characterization, see Supplementary Methods.

### Fabrication of PBO/MXene nanocomposite film

MXene powders were added to the mixture of 3.25 g MSA and 3.25 g TFA, and sonicated for 30 min to obtain MXene dispersion. Then, 3.5 g PBO nanofibre acid sol was added into MXene dispersion and sonicated for 15 min. PBO/MXene acid sol was obtained after further mixing with a planetary mixer at 2500 rpm for 10 min. Mixture of MSA/EA/H$_2$O (7.66 g/2.33 g/0.30 g) was added dropwise to the acid sol under stirring. Notably, the mixture not only dilutes the PBO/MXene acid sol to reduce its viscosity, but also triggers the gelation of the diluted sol. The sol was poured into a mold and left to stand for 24 h to obtain the acid gel. Then, the PBO/MXene acid gels were soaked in MSA/H$_2$O (170/30) and MSA/H$_2$O (140/60) for 12 h, respectively. PBO/MXene alcohol gel was obtained by replacing the acid gel with isopropanol for 24 h. Finally, the alcohol gel was dried in air to obtain PBO/MXene nanocomposite films. In the control experiments, PBO film was fabricated by a similar sol-gel-film conversion method, MXene film was prepared by vacuum-assisted filtration of dispersion of MXene nanosheets and drying in a vacuum

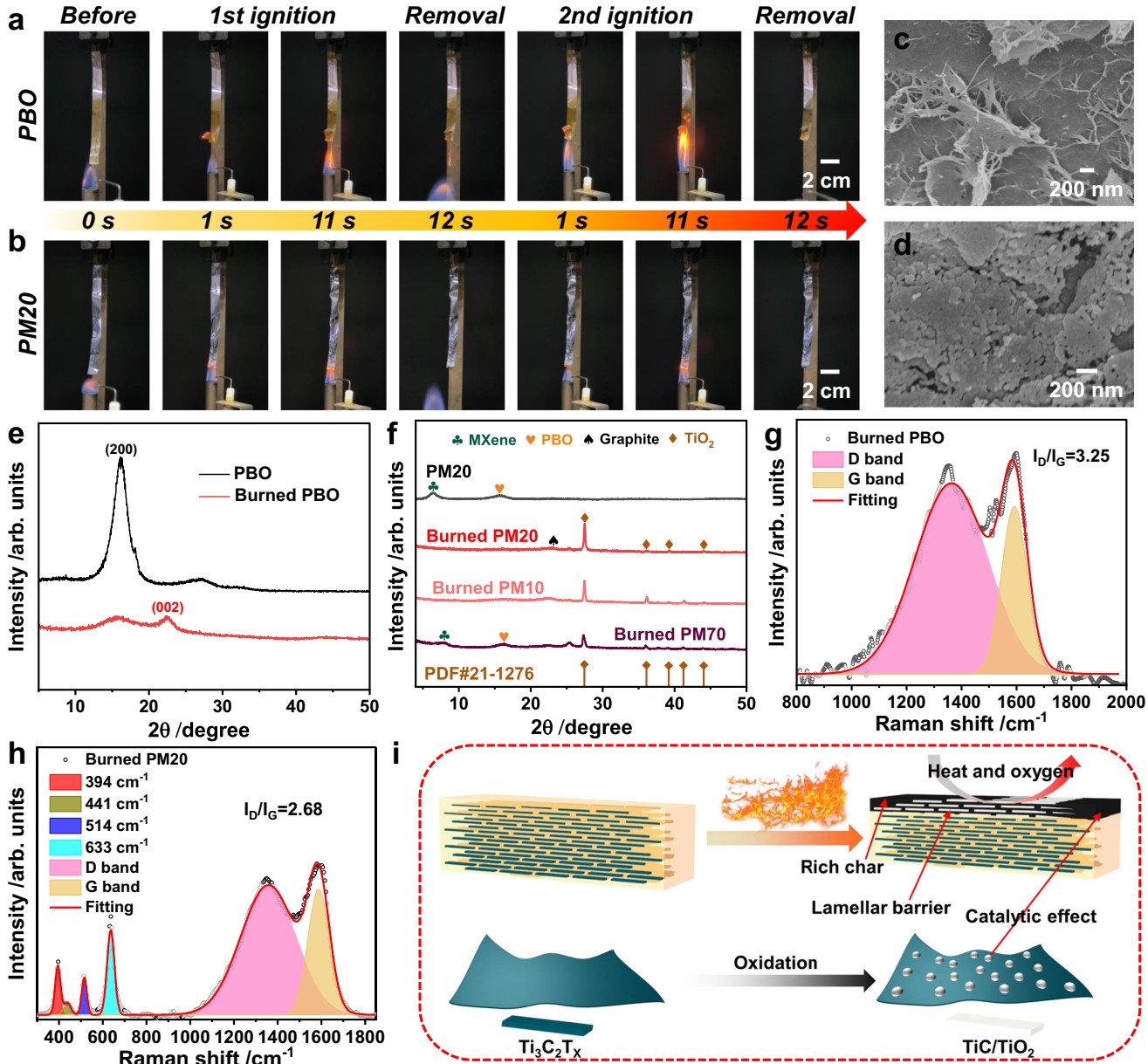

**Fig. 4 | Thermal stability and flame retardancy of PBO/MXene nanocomposite films. a, b** Snapshots of vertical combustion tests for **a** PBO and **b** PM20 films. **c, d** Surface SEM images of **c** the burned PBO and **d** burned PM20 films. **e** XRD patterns of the original PBO and burned PBO films. **f** XRD patterns of the original PM20, and burned PBO/MXene composite films (combustion time of 120 s). **g, h** Raman spectra of the burned residues of **g** PBO and **h** PM20 films. **i** Illustration for the mechanism of thermally stable and flame retardant PBO/MXene film; MXene nanosheets (bottom left) form TiO$_2$ upon burning (bottom right). The composite films have excellent thermal stability and flame retardancy owing to the shielding of heat and oxygen by the rich char and nacre-like lamellar barrier, and the catalytic effect of TiO$_2$.

oven, PBO/MXene nanocomposite film with 50 wt% MXene content was obtained by water-vapor-induced gelation (named W-PM50 film, the steps were identical except that the PBO/MXene sol was gelated by subjecting it to an environment with a relative humidity of ~99%). To characterize the morphology of the gel network by SEM, the gels were replaced with phenol in an oven at 60 °C for 48 h, and then the oven was turned off to allow the gels to freeze naturally. Lyophilized gels were then obtained by freeze-drying the frozen gels for further characterization.

### Reporting summary
Further information on research design is available in the Nature Portfolio Reporting Summary linked to this article.

### Data availability
The authors declare that data supporting the findings of this study are available within the paper and its Supplementary Information Files. Data are also available from the corresponding author upon request. Source data are provided in this paper.

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

## Acknowledgements
We are grateful for the financial support from the Chinese Academy of Sciences (Grant No. QYZDJ-SSW-SLH032, J.X.), and the National Natural Science Foundation of China (51733008, J.X.).

## Author contributions
Y.L. and N.Z. designed the experiment. Y.L. and N.Z. analyzed the data and wrote the manuscript. Y.L. prepared and characterized the materials. W.Z. helped with materials testing. N.Z. and J.X. supervised the whole process.

## Competing interests
The authors declare no competing interests.
