## [Peer Review File · Nature Communications]

REVIEWER COMMENTS

Reviewer #1 (Remarks to the Author):

This manuscript describes an interesting composite sample with high thermal conductivity and low electric conductivity, which may find applications in micro-electronics. Both of MXene and PBO are outstanding engineering materials, thus their composites are also promising for proper functions. A few problems are noticed, which should be corrected before acceptance for publication.

1. The authors claimed "The remarkably improved TC is closely related to the strong interaction between MXene nanosheets and PBO nanofibres.", and later claimed "in the vertical direction, heat conduction becomes very slow due to the layered structure with weak interactions.". Would you clarify the "weak interactions" and make it compatible to "strong interaction between MXene and PBO"?

2. The authors claimed "Fig. 4a, no flames were observed throughout the combustion of both PBO film and PBO/MXene film on an alcohol lamp.". It is not clear and does not agree with the photos. Maybe it is suitable to describe "they self-extinguished after withdrawing from flame".

3. When they discuss on the flame retardancy, it is some superficial to show only the photos of burnt samples. Standard tests are needed to prove the concept of flame retardancy, e.g., Limited Oxygen Index and vertical burning test. For details, please refer to references: ACS Appl Mater Interfaces, 2021, 13, 21876-87. J Hazard Mater 2021, 401, 123342.

Reviewer #2 (Remarks to the Author):

The paper has been well-written and can be accepted for publication in this journal after addressing following minor issues:

1- The SEM image of Ti₃AlC₂ MAX phase can be also interesting for readers, so it is suggested to add it in the revised version.

2- The fracture analyses of the PBO/MXenes using SEM images of other samples such as PB10, and PB70 and comparison between them is necessary.

3- All results show that the optimum content of Ti₃C₂ MXene in fabricated composites is 20 wt.%. However, there is not in-depth comparisons and discussions between the samples with different content of MXenes. The authors should provide more analyses for comparison of samples in the revised version.

4- The XRD analyses for other samples such as PB10 and PB70 after burning can be also interesting.

Reviewer #3 (Remarks to the Author):

The ms describes experiments on formation of biomimetic-inspired thermally conductive composite using MXene and PBO (polymer fibers). The polymer fibers contribute to the nacre-like composite the mechanical strength, electrical insulation, and fire retardancy, while the MXene particles contribute the thermal conductivity. Both fiber preparation and MXene exfoliation are described in detail, as well as the sol-gel process, solvent exchange and the formation of the composite. The ms cleverly encompasses intricate preparations with engineering demands.

The TC enhancement is strongly linked to the optimal choice of polymer.

Major concerns:

1. The authors should present the industrial requirements for thermal management applications. They state that " To date, electrical resistance above $1 \times 10^9 \Omega \text{ cm}$ and thermal conductivity exceeding $10 \text{ W m}^{-1} \text{ K}^{-1}$ have yet to be realized in MXene-based composites." However, the neat PBO in plain thermal conductivity ($\sim 25.6 \text{ W/m K}$) and electrical conductivity ($\gg 1 \times 10^9 \Omega \text{ cm}$) already meets these requirements and should be properly explained why the incorporation of MXene is crucial for industrial applications.

2. In L155-157 the authors compare the mechanical stability between MXene and PBO/MXene composite film under ultrasonication, demonstrating that the latter is more stable. However, no data regarding the degradation of PBO under sonication is presented. How does the hydrophobicity of the composite (L157) protect the composite from the shear forces induced by the sonication process?

3. Fractography analysis: L165-170, crack propagation and bifurcation, along with the toughening mechanism should be clearly demonstrated in the SEM micrographs. Moreover, fracture toughness evaluation should be considered as well to demonstrate the toughening effect on crack propagation.

4. The thickness of the films compared in Fig.3 is not described, and while the out of plane TC is substantially less than the in plane TC, layer thickness affects the heat dissipation. Is there a thickness limit from which the heat dissipation efficiency difference is negligible? This issue should be clarified.

Minor comments:

5. The authors declare that "To date, electrical resistance above 10^9 Ohm cm and thermal conductivity exceeding 10 W/mk have yet to be realized in MXene-based composites." (see also L209) This should relate to documented industrial demands and literature summary.

6. In L 205 the authors describe a practical application of in-plane TC where a high in plane TC is an advantage. It is suggested to discuss also heat sink applications (e.g., TIM) where such orientation is a disadvantage (see also L237).

7. It is a pity that the nomenclature of the systems (e.g., PM* and W-PM*)-so frequently used along the ms - is buried somewhere in the supplementary file and not explained properly in the maintext.

8. The statement on BNNS-based composites in L227 should refer to the literature.

9. The experimental setup with the ceramic plate (L239) is not explained at all. How do the voltage values relate to it? This deserves elaboration. A possible analysis could include time constant calculations for the different systems.

10. The authors are also miser in elaborating on the flame retardancy experiments (neither in the main text nor in the experimental). Were these performed according to an ASTM? Which? How do they compare to other measurements? Was dripping observed?

RESPONSE TO REVIEWERS' COMMENTS

For Reviewer #1

Comments: This manuscript describes an interesting composite sample with high thermal conductivity and low electric conductivity, which may find applications in micro-electronics. Both of MXene and PBO are outstanding engineering materials, thus their composites are also promising for proper functions. A few problems are noticed, which should be corrected before acceptance for publication.

Response: We appreciate this reviewer for taking time to review our manuscript and the valuable suggestions.

Q1: The authors claimed “The remarkably improved TC is closely related to the strong interaction between MXene nanosheets and PBO nanofibres.”, and later claimed “in the vertical direction, heat conduction becomes very slow due to the layered structure with weak interactions.”. Would you clarify the “weak interactions” and make it compatible to “strong interaction between MXene and PBO”?

Response: Thanks. The two interactions have different references. For the “strong interaction between MXene nanosheets and PBO nanofibers”, the interaction is compared to the weak interfacial interaction between MXene and PBO in the control samples with unoptimized networks (for example W-PM50). In the optimized nacre-like film, the finer MXene nanosheets and PBO nanofibres enable more hydrogen bonding and van der Waals force between them, which can greatly enhance the interfacial transport of heat carriers.

In the sentence "in the vertical direction, heat conduction becomes very slow due to the layered structure with weak interactions.", the weak interaction refers to the interaction between the layers, and is compared to the covalent bonds in the molecules of oriented MXene and PBO. This great variability is the underlying cause of the anisotropy in the thermal conductivity of the films (Mater. Today 17, 163-174 (2014)). To avoid the confusion, we have made a clarification of the weak interaction in the revised

manuscript as "in the vertical direction, heat conduction becomes very slow due to the layered structure with weak interactions (relative to the covalent bonds in the building blocks along the horizontal direction)."

Q2: The authors claimed "Fig. 4a, no flames were observed throughout the combustion of both PBO film and PBO/MXene film on an alcohol lamp.". It is not clear and does not agree with the photos. Maybe it is suitable to describe "they self-extinguished after withdrawing from flame".

Response: Thanks for the helpful suggestion. In the revised manuscript, this description has been changed as "films exposed to the flame for 10 s self-extinguished immediately after withdrawing from flame".

Q3: When they discuss on the flame retardancy, it is some superficial to show only the photos of burnt samples. Standard tests are needed to prove the concept of flame retardancy, e.g., Limited Oxygen Index and vertical burning test. For details, please refer to references: ACS Appl Mater Interfaces, 2021, 13, 21876-87. J Hazard Mater 2021, 401, 123342.

Response: Thanks for the constructive suggestion. Vertical burning tests have been carried out according to ISO 9773-1998 standard (**Fig. R1**), and the limited oxygen index (LOI) of films were obtained according to ISO 4589-2 standard (**Fig. R2**). The reference (ACS Appl Mater Interfaces, 2021, 13, 21876-87) has been cited in the revised manuscript as Ref. 44. The results show that these films exposed to the flame for 10 s self-extinguished immediately after withdrawing from flame, and no molten drippings were produced. Therefore, they all passed the VTM-0 rating. In addition, all the films have LOI higher than 50% and the composite films show a slight advantage, further demonstrating that the PBO/MXene composite films are ideal for flame-retardant materials. The original photos in **Fig. 4a, b** have been replaced by the vertical burning results, and the results of other vertical burning tests and LOI data have been

added in the revised manuscript as **Supplementary Figures 13 and 14**. The following discussion was added to the revised main text: "Vertical combustion and limited oxygen index (LOI) tests were carried out to investigate the flame retardancy of the samples⁴⁴. All the films pass the VTM-0 rating (Supplementary Figure 13) and have a high LOI larger than 50% (Supplementary Figure 14). Typically, PBO (LOI~53.5%) and PM20 (LOI~54.9%) films exposed to the flame for 10 s self-extinguished immediately after withdrawing from flame, and no molten drippings were produced (Fig. 4a, b)." (Page 16, Lines 277-282)

Fig. R1 (Supplementary Figure 13 in the revised supplementary information).
Snapshots of vertical combustion tests for PBO and PBO/MXene films.

Fig. R2 (Supplementary Figure 14 in the revised supplementary information).

Limited oxygen index of PBO and PBO/MXene films.

For Reviewer #2

Comments: The paper has been well-written and can be accepted for publication in this journal after addressing following minor issues:

Response: We appreciate this reviewer for taking time to review our manuscript, and the manuscript has been improved based on the constructive comments and suggestions.

Q1: The SEM image of Ti_3AlC_2 MAX phase can be also interesting for readers, so it is suggested to add it in the revised version.

Response: Thanks for the suggestion. We have included the SEM image of Ti_3AlC_2 MAX phase as an inset in **Fig. 1b** in the original manuscript. A red frame was added in the image to indicate the inset more clearly.

Fig. 1b in the revised manuscript. TEM image of MXene nanosheets (insets: electron diffraction pattern of MXene and SEM image of Ti_3AlC_2 MAX).

Q2: The fracture analyses of the PBO/MXenes using SEM images of other samples such as PB10, and PB70 and comparison between them is necessary.

Response: Thanks for the constructive suggestion. we have characterized SEM images of propagated cracks for other samples. As shown in **Fig. R3**, the fracture morphologies change with the MXene contents. Compared to the pure PBO film, the composite films

show significant crack bifurcation when the MXene content is in the range of 10–50 wt%. Furthermore, PM20 film has the most pronounced crack bifurcation and plastic deformation of the surrounding area, which is consistent with its highest mechanical properties. Notably, severe agglomeration of nanosheets occurred in PM70 film, leading to poor mechanical properties even below those of pristine PBO film. These results indicate that synergistic effects of the optimized nacre-like "nanosheet and nanofibre" structure can effectively inhibit the expansion of cracks and thus endow the material with strong mechanical properties. We have added SEM images of crack propagation of other samples as **Supplementary Figure 9**. The following discussion has been added to the revised manuscript: "As shown in Fig. 2g and Supplementary Figure 9, the typical "zigzag" crack propagation path, crack bifurcation and film plastic deformation can be observed during crack propagation in PBO/MXene composite films with 10–50 wt% MXene content, accompanied by slipping and pull out of MXene nanosheets, and elongation and fracture of PBO nanofibres. **Notably, more pronounced crack bifurcation and plastic deformation of the surrounding area are shown in the PM20 film compared to the other composite films, indicating a more prominent toughening effect. At MXene contents up to 70 wt%, there is a severe aggregation of nanosheets in the composite film, which could explain the poor mechanical properties even below those of the PBO film.**" (Please see Page 10)

Fig. R3 (Supplementary Figure 9 in the revised supplementary information). SEM images of the propagated cracks for PBO/MXene films with different MXene contents.

Q3: All results show that the optimum content of Ti_3C_2 MXene in fabricated composites is 20 wt.%. However, there is not in-depth comparisons and discussions between the samples with different content of MXenes. The authors should provide more analyses for comparison of samples in the revised version.

Response: Thanks for the suggestion. The SEM images of propagated cracks (Supplementary Figure 9), tensile tests of the single-edge notched films

(Supplementary Figure 10), XRD analyses (Fig. 4f in the revised manuscript), vertical combustion and limited oxygen index tests (Supplementary Figures 13 and 14) of other samples have been added. These data further confirm that an optimal nacre-like structure was formed at MXene content of 20 wt% and contributed to the superior comprehensive performances. Corresponding discussions have been added in the revised manuscript where appropriate.

Q4: The XRD analyses for other samples such as PB10 and PB70 after burning can be also interesting.

Response: Thanks for the suggestion. We have supplemented XRD patterns of PM10 and PM70 after burning (combustion time of 120 s). As shown in Fig. 4f in the revised manuscript, the XRD pattern of the burned PM10 is similar to that of burned PM20. Peaks of MXene and PBO can be observed in the burned PM70, demonstrating that the introduction of MXene is beneficial in retarding the transfer of heat and oxygen to the interior, and thus improving the thermal stability and flame retardancy of the material. The following discussion has been added to the revised manuscript: "The XRD pattern of the burned PM70 sample still shows peaks of MXene and PBO, further confirming that the introduction of MXene can improve the thermal stability and flame retardancy of the material." (Please see Page 16, lines 291-293)

Fig. 4f in the revised manuscript. XRD patterns of the original PM20, and burned PBO/MXene composite films (combustion time of 120 s).

For Reviewer #3

Comments: The ms describes experiments on formation of biomimetic-inspired thermally conductive composite using MXene and PBO (polymer fibers). The polymer fibers contribute to the nacre-like composite the mechanical strength, electrical insulation, and fire retardancy, while the MXene particles contribute the thermal conductivity. Both fiber preparation and MXene exfoliation are described in detail, as well as the sol-gel process, solvent exchange and the formation of the composite. The ms cleverly encompasses intricate preparations with engineering demands.

The TC enhancement is strongly linked to the optimal choice of polymer.

Response: We appreciate this reviewer for taking valuable time to review our manuscript. The manuscript has been revised point-by-point according to the constructive comments and suggestions.

Major concerns:

Q1: The authors should present the industrial requirements for thermal management applications. They state that “To date, electrical resistance above $1 \times 10^9 \Omega \text{ cm}$ and thermal conductivity exceeding $10 \text{ W m}^{-1} \text{ K}^{-1}$ have yet to be realized in MXene-based composites.” However, the neat PBO in plain thermal conductivity ($\sim 25.6 \text{ W/m K}$) and electrical conductivity ($\gg 1 \times 10^9 \Omega \text{ cm}$) already meets these requirements and should be properly explained why the incorporation of MXene is crucial for industrial applications.

Response: Thanks. With the rapid development of miniaturization and power density, modern electrical equipment and electronic devices place higher demands on heat dissipation. The ideal thermal management material must have good electrical insulation, but also excellent mechanical properties and flame retardancy.

Supplementary Table 1 summarizes the properties of commercially available thermal conductive plastics. It can be seen that the increase in thermal conductivity is often at the expense of electrical insulation, and the majority of thermally conductive plastics

have an electrical resistance less than $1 \times 10^9 \Omega \text{ cm}$ when the thermal conductivity is greater than $10 \text{ W m}^{-1} \text{ K}^{-1}$. Therefore, developing thermally conductive materials with a thermal conductivity exceeding $10 \text{ W m}^{-1} \text{ K}^{-1}$ and an electrical resistance above $1 \times 10^9 \Omega \text{ cm}$ remains a great challenge and has broad market demand.

Although the properties of pure PBO film already meet these criteria, the introduction of MXene not only significantly improves the thermal conductivity of the film while maintaining the electrical insulation, but also enhances the mechanical properties as well as flame retardancy. Thus, high-performance nanocomposites integrating two engineered materials, PBO and MXene, have broader prospects for industrial applications. As suggested by the reviewer, appropriate clarifications have been made in the revised manuscript: "Overall, the proper introduction of MXene not only significantly improves the TC and mechanical properties of the PBO film while maintaining the electrical insulating properties, but also further enhances the thermal stability and flame retardancy. Thus, the high-performance nanocomposite film integrating two engineered materials, PBO and MXene, have broader prospects for industrial applications.". (Please see Page 17, lines 301-305)

Supplementary Table 1 in the revised supplementary information. Properties of commercially available thermally conductive plastics.

Company	Polymer matrix	Grade	Strength (MPa)	TC ($\text{W m}^{-1} \text{ K}^{-1}$)	Electrical resistance ($\Omega \text{ cm}$)
Cool Polymers ^a	PPS	D5108	36	10	2.5×10^{16}
	PPS	D1202	25	5	$>1 \times 10^9$
	PPS	D5110	46	1.5	$>1 \times 10^9$
	LCP	D5506	55	10	1.6×10^{16}
	PPS	E5101	45	20	1.1×10^3
	PPS	E5105	60	4.5	$>1 \times 10^9$
	PA6	E3607	50	14	$<1 \times 10^9$

DSM ^a	PA46	Stanyl-TC502	65	14	1×10 ⁶
	PA46	Stanyl-TC551	50	14	1×10 ⁶
	PA46	Stanyl-TC154	55	1.0	1×10 ¹⁵
	PA46	Stanyl-TC155	55	5.0	-
	PA46	Stanyl-TC153	55	8.0	1×10 ¹⁵
Laticonther ^a	PA6	62GR/70	-	28	1×10 ²
	PPS	80GR/50	60	10	2×10 ³
	PA6	62GR/50	-	12	1×10 ⁴
Mitsubishi ^b	PC	TPN2131	81	4.9	2×10 ¹²
	PC	TPN2140	65	3.3	5×10 ¹²
	PC	TPN1125	44	21.3	2×10 ⁴
	PC	TPN1124	57	14.4	5×10 ⁴
	PC	TPN1122	80	8.8	2×10 ⁴
	PC	TPN1140	108	3.3	2×10 ⁵
	PC	TPN1022	48	13.2	2×10 ⁴
Avient ^a	PA66	NNC-5000	48.3	11	1×10 ⁴
	PA12	NJC-6000	49	11	1×10 ³
	PA12	NJC-7500	39.3	24.9	1×10 ⁵
	PA12	NJC-6500	36.4	≥16.0	2.2×10 ⁴

a: <https://www.matweb.com>.

b: <https://www.m-ep.co.jp/ch/product>.

Q2: In L155-157 the authors compare the mechanical stability between MXene and PBO/MXene composite film under ultrasonication, demonstrating that the latter is more stable. However, no data regarding the degradation of PBO under sonication is presented. How does the hydrophobicity of the composite (L157) protect the composite from the shear forces induced by the sonication process?

Response: Thanks for the question. The structural stability of the PBO film against ultrasonication was characterized. As shown in **Supplementary Figure 8**, like the PBO/MXene composite film, the PBO film was able to maintain the film integrity after 30 min of ultrasonication. This result is no doubt related to the better mechanical properties of the films themselves. As far as hydrophobicity is concerned, the hydrophobicity of PBO film and PBO/MXene composite film makes water difficult to enter into the film interior, thus protects the film backbone from being weakened by

water and mitigates the shear damage to the film caused by the cavitation bubbles generated by ultrasonication. In the revised manuscript, we added the explanation in the caption of **Supplementary Figure 8** as: "The improved hydrophobicity also protects the film backbone from being weakened by water, and mitigates the shear damage to the film caused by the cavitation bubbles generated by ultrasonication."

Supplementary Figure 8 in the revised supplementary information. **a** Water contact angle (CA) of MXene, PBO and PM20 films. **b** Optical photographs of MXene, PBO and PM20 films before and after ultrasonic treatment.

Q3: Fractography analysis: L165-170, crack propagation and bifurcation, along with the toughening mechanism should be clearly demonstrated in the SEM micrographs. Moreover, fracture toughness evaluation should be considered as well to demonstrate the toughening effect on crack propagation.

Response: Thanks for the constructive suggestion. We have clearly marked the zigzagging crack propagation path, crack bifurcation, and film plastic deformation in

the SEM micrographs (**Fig. 2g in the revised manuscript**).

In addition, tensile tests were carried out on the single-edge notched samples with notched length of 1 mm to test the fracture energy (**Fig. R4**). The fracture energy was calculated as $G_c = 6wc/\sqrt{\lambda_c}$, where λ_c is the fracture strain of the notched sample, c is the notched length and w is the energy calculated by integrating the stress–strain curve of the unnotched sample until λ_c . As shown in **Fig. R5**, the nanocomposite films exhibit a significant increase in fracture energy at MXene contents in the range of 20–50 wt% (especially 20 wt%) compared to the pure PBO film. In addition, the defects caused by the severe agglomeration of the MXene nanosheets in PM70 result in even lower fracture energy than PBO film. These comparative results indicate that synergistic effects of the optimized nacre-like "nanosheet and nanofibre" structure can effectively inhibit the expansion of cracks and thus endow the material with strong mechanical properties. The fracture energy investigation has been added as **Supplementary Figure 10** in the revised supplementary information. The following discussion has been added to the revised main text: "In addition, tensile tests of the single-edge notched films were used to calculate their fracture energy (Supplementary Figure 10). As expected, PM20 exhibits the greatest fracture energy, further demonstrating its powerful toughening effect on crack propagation." (Please see Page 10, lines 178-181)

Fig. 2g in the revised manuscript. SEM images of the propagated crack for PM20 film.

Fig. R4. Dimensions of notched tensile specimens.

Fig. R5 (Supplementary Figure 10 in the revised supplementary information). a-f Stress–strain curves of notched samples. g Statistics of fracture energy.

Q4: The thickness of the films compared in Fig.3 is not described, and while the out of plain TC is substantially less than the in-plane TC, layer thickness affects the heat

dissipation. Is there a thickness limit from which the heat dissipation efficiency difference is negligible? This issue should be clarified.

Response: Thanks for raising this issue. The thickness of the films compared in Fig.3 is about 8 μm . Typically, a much smaller thermal gradient will be established across the thickness than in the plane of the film because of the large diameter-to-thickness ratio of the film prepared (Adv. Mater. 26, 4521-4526 (2014)). Furthermore, the size of thermal management material is usually much larger than the size of the heat spots (local heat source). These facts mean that the out-of-plane TC of the film in the thickness direction nearly does not contribute to cooling, whereas the in-plane TC plays a key role in the efficient heat transfer. This has also been confirmed by many reported anisotropic thermal conductive films (Adv. Mater. 32, 1906939 (2020); Adv. Funct. Mater. 32, 2110782 (2021); J. Mater. Chem. A 9, 8527-8540 (2021); Chem. Eng. J. 437, 135482 (2022)). Among them, Song *et al.* compared highly anisotropic 15P@G-PVA/10G composite film (in-plane TC of $82.4 \text{ W m}^{-1} \text{ K}^{-1}$ and out-of-plane TC of $1.45 \text{ W m}^{-1} \text{ K}^{-1}$) with isotropic tinfoil (TC of $67 \text{ W m}^{-1} \text{ K}^{-1}$) for heat dissipation ability. Both simulation and experiment results show that anisotropic composite film can remove accumulated heat more quickly than isotropic tinfoil (**Fig. R6 and Fig. R7**, Adv. Funct. Mater. 32, 2110782 (2021)). In this study, the diameter of the films is about 65 mm, which is much larger than the thickness. In our case when films with large diameter-to-thickness ratios are used as thermal management materials for the local heat source, the contribution of out-of-plane TC to heat dissipation in the thickness direction is negligible, and the ability to eliminate heat spots depends on the in-plane TC of the film (**Fig. R8**). The corresponding clarifications have been added in the revised manuscript: "With the superior in-plane TC of PBO/MXene film in mind, we investigated its effect on dissipating heat from the local heat source (Fig. 3f, g and Supplementary Figure 12). It should be mentioned that the thickness is much smaller than the diameter for the films (PBO and PM20), and thus the contribution of out-of-plane TC to heat dissipation in the thickness direction is negligible^{39, 43}." (Please see Page 14, lines 255-258.)

Fig. R6. Modeling and calculation of the temperature for highly anisotropic 15P@G-PVA/10G composite films (in-plane TC of 82.4 W/mK, out-of-plane TC of 1.45 W/mK) and isotropic tinfoil (TC of 67 W/mK) (Adv. Funct. Mater. 32, 2110782 (2021)).

Fig. R7. Thermal infrared images of anisotropic composite film and isotropic tinfoil as

thermal management materials for LED chip (Adv. Funct. Mater. 32, 2110782 (2021)).

Fig. R8. Schematic diagram of the thermal management mechanism of PBO and PM20 films.

Minor comments:

Q5: The authors declare that “To date, electrical resistance above 10^9 Ohm cm and thermal conductivity exceeding 10 W/mK have yet to be realized in MXene-based composites.” (see also L209) This should relate to documented industrial demands and literature summary.

Response: Thanks for the helpful suggestion. In the original version, we have listed a literature summary of MXene-based thermally conductive composites. As shown in **Supplementary Table 5**, these reported MXene-based thermal conductive composites have not yet achieved both electrical insulation ($>1 \times 10^9 \Omega \text{ cm}$) and thermal conductivity exceeding $10 \text{ W m}^{-1} \text{ K}^{-1}$. Meanwhile, to better support this statement, we have summarized commercially available thermal conductive plastics in the revised supplementary information (**Supplementary Table 1**). It can be seen that the majority of thermally conductive plastics have a thermal conductivity less than $10 \text{ W m}^{-1} \text{ K}^{-1}$ when electrically insulated. The statement was changed to "To date, electrical resistance above $1 \times 10^9 \Omega \text{ cm}$ and thermal conductivity exceeding $10 \text{ W m}^{-1} \text{ K}^{-1}$ have yet to be realized in MXene-based composites (**Supplementary Tables 1 and 5**)."

Supplementary Table 1 in the revised supplementary information. Properties of commercially available thermally conductive plastics.

Company	Polymer matrix	Grade	Strength (MPa)	TC ($\text{W m}^{-1} \text{ K}^{-1}$)	Electrical resistance ($\Omega \text{ cm}$)
Cool Polymers ^a	PPS	D5108	36	10	2.5×10^{16}
	PPS	D1202	25	5	$>1 \times 10^9$
	PPS	D5110	46	1.5	$>1 \times 10^9$
	LCP	D5506	55	10	1.6×10^{16}
	PPS	E5101	45	20	1.1×10^3
	PPS	E5105	60	4.5	$>1 \times 10^9$
	PA6	E3607	50	14	$<1 \times 10^9$

DSM ^a	PA46	Stanyl-TC502	65	14	1×10 ⁶
	PA46	Stanyl-TC551	50	14	1×10 ⁶
	PA46	Stanyl-TC154	55	1.0	1×10 ¹⁵
	PA46	Stanyl-TC155	55	5.0	-
	PA46	Stanyl-TC153	55	8.0	1×10 ¹⁵
Laticonther ^a	PA6	62GR/70	-	28	1×10 ²
	PPS	80GR/50	60	10	2×10 ³
	PA6	62GR/50	-	12	1×10 ⁴
Mitsubishi ^b	PC	TPN2131	81	4.9	2×10 ¹²
	PC	TPN2140	65	3.3	5×10 ¹²
	PC	TPN1125	44	21.3	2×10 ⁴
	PC	TPN1124	57	14.4	5×10 ⁴
	PC	TPN1122	80	8.8	2×10 ⁴
	PC	TPN1140	108	3.3	2×10 ⁵
	PC	TPN1022	48	13.2	2×10 ⁴
Avient ^a	PA66	NNC-5000	48.3	11	1×10 ⁴
	PA12	NJC-6000	49	11	1×10 ³
	PA12	NJC-7500	39.3	24.9	1×10 ⁵
	PA12	NJC-6500	36.4	≥16.0	2.2×10 ⁴

a: <https://www.matweb.com>.

b: <https://www.m-ep.co.jp/ch/product>.

Supplementary Table 5. Comparison of the multifunction of nacre-inspired PBO/MXene films with other polymer/2D inorganic nanosheet composites.

Materials	Loading (wt%)	Tensile strength (MPa)	TC (W m ⁻¹ K ⁻¹)	Combustion property	Electrical insulation (>1×10 ⁹ Ω cm)	Ref.
PVA/MXene/Fe ₃ O ₄	40	21	2.9	-	No	19
CNF/MXene	60	114.4	14.9	Flame retardancy	-	20
PNF/MXene	70	125.1	5.82	Flame retardancy	No	21
CNF/ND/MXene	21.4	89.1	17.4	-	No	22

PVA/MXene	19.5	-	4.57	Flame retardancy	No	23
PBONF/GNS	50	192	100	Flame retardancy	No	7
NFC/GS	10	84.5	5.7	-	No	8
ANF/GF	20	188.5	48.2	-	No	9
CNF/rGO	8	314	29.5	-	No	24
CNF/GNP-g-L/D	5	111.8	9.36	-	Yes	25
Cellulose/BNNS	50	226	20.4	-	Yes	11
SiO ₂ -coated NFC/BNNS	7	166	10.9	-	Yes	12
ANF/BNNS	30	167	46.7	Flame retardancy	Yes	13
ANF/BN	50	60	64.1	-	Yes	14
PBO/MXene	10	342.2	33.0	Flame	Yes	This
	20	416.7	42.2	retardancy	Yes	Study

-: no data presented in the cited literature.

Q6: In L 205 the authors describe a practical application of in-plane TC where a high in plane TC is an advantage. It is suggested to discuss also heat sink applications (e.g., TIM) where such orientation is a disadvantage (see also L237).

Response: Thanks for the helpful suggestion. Yes, such orientation is often a disadvantage for heat sink applications (e.g., TIM). The orientation would be unfavorable for heat dissipation from a large surface heat source with a size larger than or close to the thermally conductive material. However, as the response to **Q4**, such orientation would significantly improve the efficiency of heat dissipation for a local heat source with a size much smaller than the thermally conductive material. Therefore, in the revised manuscript, this description has been corrected to "which contributes to

the efficiency of heat dissipation **for local heat sources** in practical applications".

Q7: It is a pity that the nomenclature of the systems (e.g., PM* and W-PM*)-so frequently used along the ms - is buried somewhere in the supplementary file and not explained properly in the maintext.

Response: Thanks. We have explained the nomenclature of the systems properly in the revised main text. (Please see Page 6, lines 92, 93, 111 and 112)

Q8: The statement on BNNS-based composites in L227 should refer to the literature.

Response: Thanks for the helpful suggestion. In the revised manuscript, the statement has referred to the relevant literature (J. Mater. Chem. A 9, 10304-10315 (2021); Composites, Part A 143, 106261 (2021); Adv. Mater. 32, 1906939 (2020); Compos. Sci. Technol. 189, 108021 (2020)). The statement was changed to "the tensile strength of our films has surpassed that of the reported BNNS-based thermal conductive materials^{10, 40-42}"

Q9: The experimental setup with the ceramic plate (L239) is not explained at all. How do the voltage values relate to it? This deserves elaboration. A possible analysis could include time constant calculations for the different systems.

Response: We are sorry for the inadequate explanation of the experimental setup. We did not constantly change the voltage of the ceramic plate. The statement "By regulating the voltage (on to off, 12 V to 0 V)" means the ceramic plate can be heated up to 228.2°C in 80 s in air by applying a fixed voltage of 12 V, then the plate will cool naturally by turning off the power supply (0 V). We have revised the statement to "**By turning the power supply on/off**". Meanwhile, a detailed schematic of the experimental setup has been provided as **Supplementary Figure 12** in the revised supplementary information.

Supplementary Figure 12 in the revised supplementary information. The setup design of tested sample as a heat spreader for high-power ceramic plate.

Q10: The authors are also miser in elaborating on the flame retardancy experiments (neither in the main text nor in the experimental). Were these performed according to an ASTM? Which? How do they compare to other measurements? Was dripping observed?

Response: Thanks for raising this issue. Vertical burn tests have been carried out referring to ISO 9773-1998 standard. As shown in **Fig. R1 (Supplementary Figure 13)**, these films exposed to the flame for 10 s self-extinguished immediately after withdrawing from flame, and no molten drippings were produced. Therefore, they all passed the VTM-0 rating. In addition, we also tested the limited oxygen index (LOI) of films according to ISO 4589-2 standard. As shown in **Fig. R2 (Supplementary Figure 14)**, these films have LOI higher than 50% and the composite films show a slight advantage. The flame retardancy experiments and related discussions have been added in the revised version: "Vertical combustion and limited oxygen index (LOI) tests were carried out to investigate the flame retardancy of the samples⁴⁴. All the films pass the VTM-0 rating (Supplementary Figure 13) and have a high LOI larger than 50% (Supplementary Figure 14). Typically, PBO (LOI~53.5%) and PM20 (LOI~54.9%) films exposed to the flame for 10 s self-extinguished immediately after withdrawing

from flame, and no molten drippings were produced (Fig. 4a, b)." (Please see Page 16, lines 277-282)

Fig. R1 (Supplementary Figure 13 in the revised supplementary information).

Snapshots of vertical combustion tests for PBO and PBO/MXene films.

Fig. R2 (Supplementary Figure 14 in the revised supplementary information).

Limited oxygen index of PBO and PBO/MXene films.

REVIEWERS' COMMENTS

Reviewer #1 (Remarks to the Author):

The MS has been well revised and become acceptable now.

Reviewer #3 (Remarks to the Author):

I read (with great interest) the detailed response of the authors to the points I raised. With the additional background information, experimental data and analyses provided, I support publication of the ms in the present form.

Following the editor's request, I read the full replies of the authors to points raised by reviewer 2. The reviewers conducted more experiments and fully answered these points. I see no reason to delay the publication of this ms.

Point-by-point response to the reviewers' comments

For Reviewer #1

Comments: The MS has been well revised and become acceptable now.

Response: We would like to thank the reviewer for recognition and recommendation of our work.

For Reviewer #3

Comments: I read (with great interest) the detailed response of the authors to the points I raised. With the additional background information, experimental data and analyses provided, I support publication of the ms in the present form.

Following the editor's request, I read the full replies of the authors to points raised by reviewer 2. The reviewers conducted more experiments and fully answered these points. I see no reason to delay the publication of this ms.

Response: We would like to thank the reviewer for the positive feedback. We truly believe that the reviewers' comments/suggestions improved significantly our work.